# *Lactobacillus* sp. Facilitate the Repair of DNA Damage Caused by Bile-Induced Reactive Oxygen Species in Experimental Models of Gastroesophageal Reflux Disease

**DOI:** 10.3390/antiox12071314

**Published:** 2023-06-21

**Authors:** Joshua N. Bernard, Vikram Chinnaiyan, Jasmine Almeda, Alma Catala-Valentin, Claudia D. Andl

**Affiliations:** Burnett School of Biomedical Sciences, College of Medicine, University of Central Florida, Orlando, FL 32827, USA; joshua.bernard@ucf.edu (J.N.B.); jasminealmeda8@knights.ucf.edu (J.A.); alma.catala@knights.ucf.edu (A.C.-V.)

**Keywords:** gastroesophageal reflux disease (GERD), Barrett’s esophagus, probiotics, reactive oxygen species, DNA damage, inflammation

## Abstract

Gastroesophageal reflux disease (GERD) leads to the accumulation of bile-induced reactive oxygen species and oxidative stress in esophageal tissues, causing inflammation and DNA damage. The progression sequence from healthy esophagus to GERD and eventually cancer is associated with a microbiome shift. *Lactobacillus* species are commensal organisms known for their probiotic and antioxidant characteristics in the healthy esophagus. This prompted us to investigate how *Lactobacilli* survive in a bile-rich environment during GERD, and to identify their interaction with the bile-injured esophageal cells. To model human reflux conditions, we exposed three *Lactobacillus* species (*L. acidophilus*, *L. plantarum*, and *L. fermentum*) to bile. All species were tolerant to bile possibly enabling them to colonize the esophageal epithelium under GERD conditions. Next, we assessed the antioxidant potential of *Lactobacilli* and role in bile injury repair: we measured bile-induced DNA damage using the ROS marker 8-oxo guanine and COMET assay. Lactobacillus addition after bile injury accelerated repair of bile-induced DNA damage through recruitment of pH2AX/RAD51 and reduced NFκB-associated inflammation in esophageal cells. This study demonstrated anti-genotoxic and anti-inflammatory effects of *Lactobacilli*, making them of significant interest in the prevention of Barrett’s esophagus and esophageal adenocarcinoma in patients with GERD.

## 1. Introduction

The number for patients suffering from gastroesophageal reflux disease (GERD), a condition of retrograde flow of stomach contents into the esophagus, are rising. In 2019, 785 million cases of GERD were reported globally, a staggering increase of about 75% from 1990 [1]. Within the US, it is one of the most diagnosed disorders of the gastrointestinal tract. A prevalence of about 20% is a significant economic and health care burden in addition to negatively affecting the quality of life of those suffering from GERD [2]. Exposure of the esophagus to stomach refluxate, which contains gastric acids as well as bile salts, is known to cause cellular and DNA damage leading to inflammation and, if the condition persists, to Barrett’s esophagus, a premalignant condition, and the main risk factor for esophageal adenocarcinoma [3,4,5]. The link between even short exposures to low pH and bile acid and oxidative stress and oxidative DNA damage was recognized early [5]. Repeated DNA damage due to bile exposure is thought to increase the mutation rate, including mutations in tumor suppressor genes and oncogenes [6].

More recently, probiotics were proposed to be alternative treatments in the prevention of cancers. Probiotic lactic bacteria, especially, are considered to be important microbes with anti-carcinogenic activities [7]. Different *Lactobacillus* strains show anti-proliferative properties in vitro and in vivo [8]. *Lactobacilli* are important commensals, and most strains are acid tolerant, which allows them to colonize certain niche environments such as the gastrointestinal tract unless they are exposed to acids for prolonged times, causing growth inhibition or cell death [9]. We aimed to evaluate the adaptation of different *Lactobacillus* strains under conditions of bile salt exposure to mimic the esophageal environment of GERD. Bile functions as an emulsifier for lipids and can also dissolve bacterial membranes, thereby conferring anti-microbial activity [10]. Primary bile acids such as cholic and deoxycholic acids are synthesized in the liver and conjugated with glycine or taurine prior to secretion to aid in their solubility. Deconjugation is catalyzed by bile salt hydrolase enzymes, which are expressed by several *Lactobacillus* strains, to hydrolyze the amide bond and separate the glycine or taurine [10]. Bacterial strains with bile salt hydrolase activity can, therefore, be found in the gastrointestinal tract, and we hypothesized that they can colonize the esophagus under GERD conditions in which there is an enrichment in conjugated and unconjugated bile acids. The progression from the normal squamous esophagus to GERD and Barrett’s Esophagus (BE) correlates with changes in the microbiome, mostly from resident Gram-positive bacteria, to a community composed of Gram-negative bacteria [11]. Interestingly, while *Lactobacillus* sp. seem less frequent in the GI tract [12], we proposed that the diseased esophagus may provide a specific niche for it. It was shown that *Lactobacilli* abundance increases in the progression to esophageal adenocarcinoma by 16S ribosomal RNA sequence analysis [13]. Given their antioxidant effects by scavenging hydroxyl radicals and superoxide degradation amongst others [14], we interrogated the effects of *Lactobacilli* interaction with esophageal cells to determine their contributions to the repair of DNA damage induced by reactive oxygen species associated with bile injury.

In this study, we utilized three *Lactobacillus* species, *L. acidophilus*, *L. plantarum*, *L. fermentum*, and showed their adaptation to two unconjugated bile salts (sodium cholate and sodium deoxycholate) and ox bile. We assessed the interaction of *Lactobacilli* and esophageal cells in the GERD-associated genotoxic environment in vitro, i.e., bile exposure, and investigated if oxidative stress and DNA damage can be corrected by the probiotic bacteria. We found that *Lactobacilli* added after bile injury and subsequent DNA damage, as indicated by the presence of 8-oxo-guanine and the COMET assay, reduced NFκB-mediated inflammation. They can also accelerate the recruitment of double strand DNA repair proteins, such as nuclear pH2AX and RAD51, indicating an accelerated DNA damage response in the presence of *Lactobacilli*.

## 2. Materials and Methods

### 2.1. Cell Lines

As described previously [15], human squamous esophageal epithelial cells (STR) were cultured in Keratinocyte Serum-Free Medium (Gibco^TM^ Life Technologies Co., catalogue number #10724-011, Grand Island, NY, USA), supplemented with Epithelial Growth Factor (1 ng/mL), Bovine Pituitary Extract (0.05 mg/mL), and 1% Penicillin/Streptomycin (Gibco^TM^ Life Technologies Co., #15140-122, Carlsbad, CA, USA). Cells were incubated at 37 °C with 5% CO_2_.

### 2.2. Bacterial Cultures

*Lactobacillus acidophilus* (#4356), *Lactobacillus plantarum* (#14917), and *Lactobacillus fermentum* (#14931, all American Type Culture Collection, ATCC, Manassas, VA, USA) were all grown in De Man, Rogosa, and Sharpe broth (MRS; Becton Dickinson, #288130, Sparks, MD, USA) at 37 °C and 5% CO_2_. For standard serial dilution plating, bacteria were grown on MRS plates. Bacteria were sub-cultured in MRS from an independent colony, snap-frozen in liquid nitrogen for 30 min during the exponential growth phase, and stored at -80 °C. To determine CFU/mL, bacteria were serially diluted in PBS and spot plated (10 µL) on MRS plates.

### 2.3. Adaptation Assay

To determine the adaptation ability of *Lactobacilli* to bile, we carried out adaptation assays as previously described by Begley [16] with slight modifications: *Lactobacillus* species were exposed to bile salts with a 1:1 ratio sodium cholate/sodium deoxycholate (CA/DCA) (Sigma-Aldrich, #B8756, St. Louis, MO, USA) at a 1% (positive control), 0.30% (“lethal dosage”) and 0.08% (“sub lethal dosage”) concentration during exponential growth phase for 30 s as indicated [16]. For adaptation, *Lactobacillus* species were treated with 0.08% “sub lethal dosage” for either 5 s, 30 s, 5 min, or 30 min to adapt prior to being exposed to the 0.30% “lethal dosage” for an additional 30 s. Cultures were then serially diluted in PBS and spot plated (10 µL) on MRS plates and CFUs/mL were obtained.

### 2.4. Bacterial Growth Curves

Overnight cultures were diluted 1:10 in MRS and loaded into a 96-well plate (100 µL per well). The CA/DCA bacteria were pre-exposed for 30 s to CA/DCA bile and washed 3 times with PBS prior to inoculation in MRS. For experiments using ox bile (Sigma-Aldrich, #70168, St. Louis, MO, USA), the bacterial cells were inoculated in pre-treated MRS at various concentrations. Then, 96-well plates were incubated for 24 h at 37 °C in a Biotek Synergy plate reader (Agilent, Santa Clara, CA, USA). During this incubation, optical density (600 nm) readings were taken every 5 min. Growth curve data were plotted and analyzed on GraphPad (Prism 8). These experiments were carried out in technical triplicates and repeated twice.

### 2.5. Biofilm

Overnight cultures were diluted 1:10 and incubated for 24 h in a 96-well plate (100 μL per well). Bacterial cells were inoculated in pre-treated MRS supplemented with either CA/DCA or ox bile. After 24 h of incubation, plates were washed with deionized water and air dried at 37 °C for 20 min. To stain the biofilm, safranin #65092B-95 purchased from Millipore Sigma (Darmstadt, Germany) was prepared (125 mL containing 6.04 g/L safranin, 19% ethanol, and 1% methanol, diluted in water) and added to the wells. After a 20 min incubation at room temperature, the safranin was discarded, and the wells were washed with deionized water and air dried at 37 °C. The stained biofilm was dissociated by resuspending in ethanol–acetone mix (80:20). The absorbance was read at OD490 nm in a Biotek Synergy plate reader (Agilent, Santa Clara, CA, USA). Results were normalized to blanks containing media only, graphed, and analyzed using GraphPad Prism (Prism 8). The experiments were performed as technical triplicates and in 3 biological replicates.

### 2.6. Hydrophobicity

To measure bacterial surface hydrophobicity, we adapted the method from Serebryakova with slight modifications [17]. Bacterial cultures post bile exposure at stationary phase (2 mL) were mixed with chloroform (500 mL), vortexed for 2 min, and were then incubated at 37 °C for 15 min to allow separation between the aqueous and hydrophobic phases. During this separation, the optical density of the aqueous (hydrophilic) phase decreases as the hydrophobic bacteria moved into the chloroform (hydrophobic) phase. The aqueous layer (*A_a_*) was collected, and OD600 nm was measured. The OD600 nm of the overnight culture (*A_t_*) was measured to account for the total bacteria before the phase separation. The percent hydrophobicity was calculated as follows:$$Hydrophobicity\%=\left[\left(A_{t}-A_{a}/A_{t}\right)\right]\times 100$$

### 2.7. Esophageal Epithelial Cell Treatment with Bile and/or Lactobacillus

Two different bile exposures were used to mimic bile reflux. Ox bile (dehydrated and purified) (Sigma-Aldrich, #70168, St. Louis, MO, USA) and sodium cholate/deoxycholic acid sodium salt mixture at 1:1 (Sigma-Aldrich, #B8756, St. Louis, MO, USA). A 15% ox bile stock was prepared using Keratinocyte Serum-Free Media (Life Technologies Co., 10724-011, Grand Island, NY, USA) without the addition of penicillin/streptomycin, and a 10% CA/DCA stock was prepared in the same media. During bile treatment, epithelial cells were exposed to 0.30% ox bile or 0.08% CA/DCA for 30 s to simulate reflux. The cells were then washed three times with PBS and new Keratinocyte Serum-Free Media was added. During the “recovery” conditions, unexposed *Lactobacilli* were added to epithelial cells at a multiplicity of infection (MOI) of 1 after the epithelial cells were exposed to either ox or CA/DCA bile for 30 s. Then, the cells were incubated for a 30 min “recovery” in media (with bacteria or media only control) following the treatments before analysis with the COMET Assay or Immunofluorescence staining.

### 2.8. COMET Assay

We performed the COMET Assay to determine DNA damage (Abcam, Waltham, MA, USA) as per manufacturer’s instructions, utilizing the TBE Electrophoresis running solution for neutral conditions. Epithelial cells were treated with bile, *Lactobacilli* or both as explained above (Section 2.7). The length of the COMET was determined by measuring from the middle of the head to the end of the tail.

### 2.9. Immunofluorescence

Human epithelial cells were seeded (50,000 cells/well) in chamber slides (Nalge Nunc International, #154526, Rochester, NY, USA), and treated with bile, *Lactobacilli*, or both, as explained above (Section 2.7). Cells were incubated for 30 min following treatment. Next, cells were washed with PBS and fixed with methanol/acetone (1:1) for 10 min at room temperature. After fixation, cells were permeabilized with 0.5% Triton X-100 for 20 min and blocked with 1% BSA for 1 h. Slides were incubated with primary antibody overnight at 4 °C (pH2A-X 1: 400; NFκB 1: 400; both purchased from Cell Signaling Technology (Danvers, MA, USA); RAD51 (Abcam, Waltham, MA, USA) at 1: 200; 8-oxoguanine (Millipore,1: 250), washed with PBS, and incubated with corresponding secondary antibody (Alexa Fluor 488 goat anti-mouse, Invitrogen by ThermoFisher A11001, Eugene, OR, USA; Alexa Fluor 594 goat anti-mouse, Invitrogen by ThermoFisher A11001, Eugene, OR, USA) overnight at 4 °C. Slides were mounted with DAPI (Southern Biotech 0100-20, Roskilde, Denmark).

### 2.10. Statistical Analysis

Experiments were performed in biological replicates, each consisting of at least three technical replicates. The data are presented as mean ± standard error of the mean (SEM; *n* = 3). When comparing only two conditions to one another, we used a Student’s *t*-test (Welch’s correction) to analyze statistical significance differences. Statistical significance was considered when * *p* < 0.05, ** *p* < 0.01, *** *p* < 0.001, **** *p* < 0.0001. When comparing multiple conditions, we used a one-Way ANOVA. A statistically significant ANOVA was followed by Dunnett’s correction and multiple groups were compared to control. Statistical significance by Dunnett’s correction was considered when * *p* < 0.05, ** *p* < 0.01, *** *p* < 0.001, **** *p* < 0.0001.

## 3. Results

### 3.1. Lactobacilli Adapt to Cholate/Deoxycholate Bile Salt Challenge

Enteric bacteria are able to resist the high concentrations of bile encountered throughout the gastrointestinal tract, as shown for *Salmonella sp., Escherichia coli, Bacillus cereus*, and *Listeria monocytogenes* [18]. Although little information is available on the bile tolerance of Gram-negative bacteria, it is believed that they are inherently more resistant to bile than Gram-positive bacteria. Bile salts are, therefore, often used in the selection of Gram-negative bacteria in laboratory culture conditions, e.g., MacConkey Agar [19].

Deoxycholic acid is a metabolic product of gut bacteria utilizing cholic acid, and it is thought to be especially toxic due to its hydrophobic nature. We observed that compared to a 1% dose used as a control for killing all three species, 0.30% induced bacterial cell death in *L. plantarum* and *L. fermentum* and reduced cell viability for *L. acidophilus* significantly (Figure 1a–c), demonstrating that *Lactobacilli* only have limited resistance to bile at these concentrations. However, exposure to 0.08% had little effect on viability as measured by colony forming units per milliliter (CFU/mL), even though a reduction in CFUs/mL was noted for *L. acidophilus* and *L. fermentum* (Figure 1a–c). A study using Listeria [16] demonstrated possible adaptation to 0.30% bile salts, if pre-exposed to a “sub-lethal” dose 0.08%. Therefore, we designed experiments to adapt *Lactobacillus* spp. to cholate/deoxycholate and promote survival. Bacteria were treated with 0.08% bile for either 5 s, 30 s, 5 min, or 30 min before being exposed to the lethal bile concentration of 0.30% for 30 s. A 5 s pre-exposure was sufficient to adapt *L. acidophilus* and *L. fermentum*, while 30 s of 0.08% were required to ensure adaption of *L. plantarum* to cholate/deoxycholate (Figure 1d–f). These data showed that tolerance can be acquired upon exposure to lower concentrations of bile.

### 3.2. Bile Exposure Causes a Delay in Exponential Growth without the Induction of a Stress Response

We next determined the effect of 0.08% cholate/deoxycholate (CA/DCA) at 30 s pre-exposure on the recovery of bacterial growth over a period of 24 h. While we measured a delayed entry into the exponential phase in all three *Lactobacilli,* the overall growth curve was not altered, and all three *Lactobacilli* reached the stationary phase comparable to the untreated control (Figure 2a). To assess if the exposure to CA/DCA induces a stress response, we performed a biofilm formation assay in which bacteria cluster by either attaching to each other or a surface. Bacterial biofilms, often composed of bacteria in a self-produced matrix, aids the survival of bacteria in diverse environmental niches [20]. No stress-induced biofilm was measured in the 24 h of 0.08% CA/DCA supplementation to the bacterial media (Figure 2b). Biofilm formation is also an important step in the pathogenic colonization of bacteria in the human host [20]. Cell surface hydrophobicity is another important property that allows bacteria to adhere to biotic and abiotic surfaces as well as the penetration of host tissues [21]. While the attachment to surfaces depends not only on the hydrophobicity of cells, hydrophobicity is another indicator of aggregation and colonization [22]. No change in hydrophobicity was noted in *L. acidophilus*, but *L. plantarum* and *L. fermentum* showed a statistically significant increase in hydrophobicity upon CA/DCA exposure (Figure 2c), indicating a potential increase in their ability to attach and colonize the host.

### 3.3. Ox Bile as a Physiological Model for Refluxate Shows a Dose Dependent Effect in Lactobacilli Growth with Complete Recovery and No Change in Biofilm Formation

The physiological composition of bile in the intestine and the gastro-esophageal refluxate both contain a complex mixture of different conjugated and unconjugated bile salts [23,24]. The range of bile acid concentrations in the human gut is between 0.2 and 2% [25]. Since ox bile and human bile have a similar bile acid composition, ox bile, at a concentration of 0.30% (*w*/*v*), was used to measure the bile tolerance of probiotic bacteria [26]. Therefore, we utilized 0.30% (*w*/*v*) ox bile to reproduce the data investigating growth (Figure 3a), biofilm formation (Figure 3b), and hydrophobicity (Figure 3c). The growth kinetics showed a lag phase of 8 h for *L. acidophilus* and 5 h for *L. plantarum*. All three *Lactobacilli* reached the stationary phase comparable to the untreated control. Furthermore, in concordance with the experiments using CA/DCA, we observed no significant differences for biofilm induction, and only *L. plantarum* responded with a significant increase in hydrophobicity to the exposure of ox bile.

### 3.4. Lactobacilli Show Anti-Inflammatory Capabilities in the Context of Bile-Induced NFκB Signaling

A link between GERD and inflammation was provided through the key transcription factor, nuclear factor kappa B (NFκB), which was reported to be activated by bile acids in human cells and contribute to esophageal cancer progression [27,28,29,30,31]. In addition, NFκB signaling activation confers apoptotic resistance in response to deoxycholate-induced DNA damage [32]. Although bile acids generally induce DNA damage, some bile acids, such as the hydrophilic ursodeoxycholic acid (UDCA), can protect from oxidative stress in Barrett’s cells exposed to acid and bile salts. Pretreatment with UDCA decreases oxidative stress, DNA damage, and NFκB activation [33]. We, therefore, determined a treatment protocol with CA/DCA and ox bile for immortalized human esophageal epithelial cells (Appendix A), demonstrating tolerance to 0.30% ox bile for up to 30 min and 0.08% CA/DCA for 30 s without reduced cell viability. In subsequent experiments, we exposed esophageal epithelial cells for 30 s to 0.08% CA/DCA and 0.30% ox bile as relevant to the physiology of GERD before a 30 min recovery period. Known for its antioxidant properties, we added *L. acidophilus*, *L. plantarum*, and *L. fermentum* during the 30 min recovery after the 30 s bile exposure and assessed the signal and translocation of NFκB to the nucleus, which indicates the activation of the signaling pathway. Our data demonstrate that exposure to CA/DCA (Figure 4a) and ox bile (Figure 4b) induces an increase in NFκB-positive human esophageal epithelial cells, indicating inflammation as reported previously [27,28,29,30,31]. Unexposed *L. acidophilus*, *L. plantarum*, and *L. fermentum* added during recovery reduced the number of NFκB-positive cells to almost the same level as bacteria alone and control cells (Figure 4).

### 3.5. Lactobacilli Show an Antioxidant Effect

Reactive oxygen species (ROS) generate various modified DNA bases. 8-oxo-7,8-dihydroguanine (8-oxo-G) is one of the most abundant and stable products resulting from reactive oxygen species modifying guanine and is highly mutagenic as it pairs with adenine when not repaired [34]. It is widely used as a marker for ROS [35]. Notably, the accumulation of ROS was shown to contribute to the neoplastic progression of the GERD-BE-EAC sequence [36]. We confirmed the generation of 8-oxo-G using detection by immunofluorescence microscopy after 30 min upon a 30 s exposure with CA/DCA and ox bile (Figure 5a,b, respectively). The data demonstrate that about 40% of cells become 8-oxo-G-positive upon treatment compared to 10–20% in untreated control conditions. When un-exposed *L. acidophilus*, *L. plantarum*, and *L. fermentum* were added during the recovery from the bile insult, the level of 8-oxo-G-positive cells returns to levels comparable to untreated control cells and bacteria-treated cells. These observations indicated a reduction in ROS-mediated DNA damage through *Lactobacilli*-mediated protective mechanisms.

### 3.6. Addition of Lactobacilli to Esophageal Cells after Exposure to Bile Reduces DNA Damage

It was reported that exposure of esophageal cells to acidic bile salts leads to the generation of superoxide radicals and hydrogen peroxide, resulting in significant DNA damage [37]. Such DNA damage is a mechanism that possibly underlies the progression from Barrett’s esophagus to esophageal adenocarcinoma. At the molecular level, DNA damage from genotoxic insults such as bile acids can be visualized through a COMET assay, a method of gel electrophoresis of nuclei in which DNA with extensive strand breaks produces a comet-like pattern [38]. We performed COMET assays 30 min after a 30 s exposure to CA/DCA or ox bile to determine the extent of severely damaged DNA as it migrates away from the undamaged DNA containing the nucleoid body during electrophoresis. We quantified the distance between the head and tail of the comet, the undamaged DNA nucleoid part and the trailing damaged DNA streak, respectively. We showed that exposure to CA/DCA (Figure 6a) and ox bile (Figure 6b) induced an increased COMET compared to untreated controls. In the presence of unexposed *L. acidophilus*, *L. plantarum*, and *L. fermentum* during recovery, the COMET was comparable in size to untreated control and bacteria alone.

### 3.7. DNA Double Strand Break Repair Is Accelerated upon Addition of Lactobacilli

While DNA damage such as 8-oxo g is repaired by base excision repair through DNA N-glycosylases such as OGG1 [39], the response to DNA damage of double stranded breaks induced by bile salts involves canonical non-homologous end joining or homologous repair. In the context of GERD, it was reported that bile acid exposure caused an up-regulation of nuclear pH2AX [40]. We, therefore, used pH2AX to evaluate DNA strand breaks caused by repair incision nucleases and RAD51 to identify the activity of repair by homologous recombination. Thirty seconds after CA/DCA (a) or ox bile (b) treatment, we allowed the cells to recover for 30 min in the presence or absence of *Lactobacilli* before we assessed the phosphorylation of H2AX (pH2AX) and RAD51 signal. The recruitment of pH2AX was described as being rapid. The phosphorylation of serine 139 on H2AX is completed within minutes after the occurrence of double-strand breaks [41]. In the context of ROS induction, this process happens within 20 min, e.g., after neocarzinostatin treatment [42].

Using immunofluorescence, we determined that both DNA repair markers were elevated upon CA/DCA (Figure 7a and Figure 8a) and ox bile treatment compared to untreated control (Figure 7b and Figure 8b). When *L. acidophilus*, *L. plantarum*, and *L. fermentum* were added during the recovery phase after a 30 s bile-mediated DNA damage induction, the number of pH2AX and RAD51-positive cells was significantly reduced and comparable to untreated control and bacteria-alone-treated cells, indicating an accelerated repair response. A Western Blot analysis of samples representing a time course of pH2AX activation after DNA damage induction highlighted a stronger signal for pH2AX 15 min after bile-induced DNA damage in the presence of unexposed *L. acidophilus* compared to ox bile alone (Appendix A), suggesting an increased recruitment of pH2AX. We believe these findings could indicate an antigenotoxic function of probiotic *Lactobacilli*.

## 4. Discussion

The current existing medical treatment for GERD mostly refers to drugs intended to suppress the production of acid, such as proton pump inhibitors (PPIs). While proton pump inhibitors provide symptomatic long-term relief, the treatment may only have limited effect on the prevention of esophageal adenocarcinoma. PPIs do not correct the underlying reflux damage in Barrett’s esophagus, and gastroesophageal reflux even of weakly acidic material during PPI treatment still contains bile salts [43,44]. The question was raised whether bile salts or bile acids are at the heart of Barrett’s metaplasia in the esophagus [45,46,47]. Bile salts are formed in the liver by conjugation of primary bile acids such as cholic and chenodeoxycholic acids with taurine and glycine, which affects their physiological properties and negative charge [48]. Secondary bile acids including deoxycholic acids are more potent inducers of intestinal metaplasia, as seen in Barrett’s esophagus, and are responsible for DNA damage [32].

### 4.1. Antioxidative Function of Lactobacilli during ROS-Induced Inflammation and DNA Damage

It was previously reported that acidic bile salts generate high levels of ROS, oxidative stress, and associated DNA damage, such as double strand breaks, in esophageal cells [5,49,50]. We, therefore, investigated whether probiotic *Lactobacillus* treatment plays a role in regulating these events and the repair of genotoxic damages. As a cellular model, we used normal esophageal cells immortalized with hTERT and studied their interaction with *Lactobacilli* as well as their response to bile salts in the presence and absence of *Lactobacilli*. DNA damage was observed in GERD patients and described to be induced by experimental bile salt reflux in in vitro studies using BE cells where an episode of reflux was recapitulated by a short exposure of bile [51]. Due to their known role in the pathogenesis of BE tumorigenesis, [20] we selected cholic and deoxycholic acid, and a physiological mixture of bile acids collected from ox bile for our study. These conditions showed the activation of NFκB -mediated inflammation, ROS, and DNA damage.

ROS can cause DNA damage, including the formation of 8-oxo-7,8-dihydro-2′-deoxyguanosine (8-oxo-G). Guanine, having the lowest redox potential, was shown to be the DNA base that was the most susceptible to oxidation, yielding products such 8-oxo-G. 8-oxo-G was shown to be mutagenic, yielding GC to TA transversions upon incorporation of dAMP opposite to the lesion by replicative DNA polymerases. During the progression of BE, increased levels of 8-oxo-G were observed [52,53]. We showed an increase in 8-oxo-G caused by bile salt exposure of esophageal cells and a reduction after the incubation with *Lactobacilli*. 8-oxo-G is primarily repaired by the base excision repair pathway initiated by DNA N-glycosylases such as OGG1. Proteins involved in DNA repair such as 8-oxo-G DNA glycosylase-1 can promote NFκB-mediated transcription of pro-inflammatory genes [54,55,56,57], creating a vicious cycle between DNA damage and inflammation, which was also described for Barrett’s esophagus where the chronic inflammation perpetuates DNA damage [58]. GERD was reported to be the result of not just direct physical injury due to acidic bile salt exposure but also the inflammation associated with the release of cytokines [59]. Acid and bile were described to induce pro-inflammatory signaling though the cytokines IL-8 and IL-1 through the activation of NFκB in Barrett’s esophagus [60,61]. DNA damage caused by bile salts is also linked to the activation of pro-inflammatory signaling, including NFκB [62] or various interleukins leading to chronic inflammation.

Reactive oxygen species play an important role in cellular signaling and can induce oxidative stress resulting in cellular damage. Amongst the mechanisms protecting from oxidative stress are enzymatic reactions catalyzed by glutathione peroxidases (GPX) and glutathione reductase, or glutathione [63,64,65]. GPXs lower H_2_O_2_, intracellular ROS, oxidative DNA damage, and double-strand breaks following exposure of esophageal cells with acidic bile salts. GPX7, for example, is frequently silenced in esophageal adenocarcinoma and impacts its capacity in regulating ROS and oxidative DNA damage [49]. In addition, loss of GPX7 promotes the activation of NFκB via TNF in pro-inflammatory signaling during Barrett’s progression [66,67].

Other redox molecules relevant to Barrett’s tumorigenesis were investigated: Apurinic/apyrimidinic endonuclease 1 (APE1) protein plays an important role in promoting esophageal cancer cell survival by counteracting the lethal effects of DNA damage induced by acidic bile salts [68], and Chen et al. discovered a novel crosstalk between the redox function of APE1, reflux-induced inflammation and NFκB-mediated upregulation of Notch signaling resulting in the promotion of cancer cell stem-like properties in response to reflux conditions [69]. Natural antioxidants, such as apocynin, were reported to be effective in protecting esophageal cells from ROS and DNA damage induced by acidic bile salts [37]. APE1-redox-dependent function may be a mechanism in the context of *Lactobacilli* treatment which will require future investigation.

Probiotics may play an important role in oxidative stress-associated DNA damage. In lactic acid bacteria, antioxidant enzymes, such as NADH oxidase, glutathione reductase, glutathione S-transferase, catalase, glutathione peroxidase, and feruloyl esterase, counteract oxidative stress [70,71]. In this context, some *Lactobacilli* and *Bifidobacteria* were reported to induce increased activity of anti-oxidative enzymes or modulate signaling resulting in protection from oxidative stress [72]. Of importance for our study is that bacteria, such as *Lactobacilli*, also release and promote the production of glutathione itself, a major cellular non-enzymatic antioxidant [73,74]. We are yet to investigate if the observed anti-inflammatory and antioxidant functions are directly mediated by *Lactobacilli*-secreted factors or if they induce a host antioxidant response. However, a recent study showed that that probiotic bacteria such as *Lactobacillus* and *Bifidobacterium* can not only modulate the activity of bile acids, e.g., through BSH, but they can also counter act the negative effect of bile, including the repair of their own bacterial DNA upon damage [75]. Some bacteria, including *Lactobacillus*, may even adapt their metabolism with the activation of glycolysis in the presence of bile [76].

### 4.2. DNA Damage Repair

As a result of mutagenic lesions such as 8-oxo-G formation, DNA damage response signaling is activated, aiming to repair the DNA lesion [77]. Bile salts can induce double stranded breaks, which are marked by pH2AX for repair, and RAD51 during homologous recombination, which we demonstrated to be recruited and accelerated in the presence of *Lactobacilli* following bile salt-induced DNA double strand breaks.

DNA double strand breaks are repaired by canonical non-homologous end joining or homologous recombination (HR) [78,79]. Double strand break repair by canonical non-homologous end joining occurs throughout the cell cycle, whereas the HR pathway functions only during the S and G2 phases because it requires a homologous DNA sequence from the sister chromatids to serve as a template for DNA-synthesis-dependent repair [80] and it involves extensive DNA-end processing. It is, therefore, considered to be extremely accurate [81].

We are yet to analyze host cell cycle entry in the presence and absence of the *Lactobacilli* to identify the role they play in DNA repair coordination. However, when a double strand break occurs, kinases such as ATM are activated immediately and phosphorylate histone H2AX at the site of the double strand break [82,83]. Formation of pH2AX foci serves as a checkpoint for both non-homologous end-joining repair and for HR [84] through mechanisms that indirectly or directly retain effector proteins [85]. One member of the histone H2A family, H2AX, became extensively phosphorylated within minutes of DNA damage and forms foci at break sites. pH2AX is necessary for the recruitment of other factors, such as RAD51, to the sites of DNA damage [86]. pH2AX was used to evaluate DNA strand breaks caused by repair incision nucleases and RAD51 was used to measure the activity of homologous recombination in the repair. We observed that the signal for pH2AX increased in protein samples within 15 min with a stronger signal in bile injured conditions and decreased by 30 min. Detection by immunofluorescence showed that the protein signal strength and number of pH2AX as well as RAD51 foci decreased within 30 min after supplementation with *Lactobacilli* to the bile-injured cells. These findings indicate that double strand break repair was underway as the gradual dephosphorylation of pH2AX occurs as repair progresses [87]. As RAD51 catalyzes core reactions of HR including strand invasion into duplex DNA, the pairing of homologous DNA strands, thereby, enables strand exchange [88], which is likely to be a HR repair process favored in the presence of *Lactobacilli*.

pH2AX was shown to be induced by acidic bile salts in GERD patients and in vitro using the BE cell lines exposed to acidic bile salts. These findings correlated with an increase in DNA damage as measured by alkaline COMET assay [89]. We reported that bile in our experimental model induced DNA damage, as measured by 8-oxo-G and the COMET assay, as well as the induction of pH2AX and RAD51.

Of note, it was shown that some Gram-negative bacteria can induce host DNA damage through toxins or metabolites secreted by the bacteria. Additionally, it was shown that when *E. coli* induces a DNA damage response in host cells, incomplete DNA repair can occur and result in anaphase bridges and chromosomal abnormalities [90,91]. Ultimately, these can result in gene mutations and malignant transformation. As of yet, we did not further assess the outcomes of the accelerated repair but recognized that this an important future direction. However, it was shown that RAD51 prevents the accumulation of replication-associated double strand breaks and genome instability that promote cell survival following replication stress [92], suggesting that the accelerated repair may not negatively impact the esophageal cells.

While we did not determine the direct mechanisms by which *Lactobacilli* enhance DNA damage repair and recovery, multiple bacterial pathways could be involved in the mediation of anti-oxidative effects. A recent study identified specific esophageal microbiome signatures in patients with BE and EAC [93]. Of interest, the microbiota associated with Barrett’s esophagus was characterized by gene signatures organized in categories such as higher potential for replication and repair, genetic information processing, and the metabolism of cofactors and vitamins among others.

*Bifidobacterium* in specific was already demonstrated to have a beneficial function in the gastrointestinal tract [94,95]. In addition, reports for *Lactobacilli* were shown to treat dyspepsia, providing a clinical context [96,97]. Probiotics such as *Lactobacillus* spp. as well as *Bifidobacterium* are of interest due to their anti-cancer effects. In the context of esophageal cancer, an increased uptake of fermented foods, including probiotics, were shown to reduce the cancer risk [98]. This highlights their potential functions even in the disease progression.

In summary, our results indicate that *Lactobacillus* plays a critical role in regulating oxidative stress and genotoxic events and can accelerate the DNA damage response to reflux conditions if utilized as a treatment.

## 5. Conclusions

We conclude that *Lactobacilli* can tolerate or adapt to bile concentrations mimicking GERD. If supplemented to the bile-injured esophageal epithelium, *Lactobacilli* have an anti-inflammatory and antioxidant effect. This activity was demonstrated by a suppression of NFkB and the rapid activation of pH2AX and RAD51. Probiotic *Lactobacilli* could, therefore, have a beneficial role in GERD.

## Figures and Tables

**Figure 1 antioxidants-12-01314-f001:**
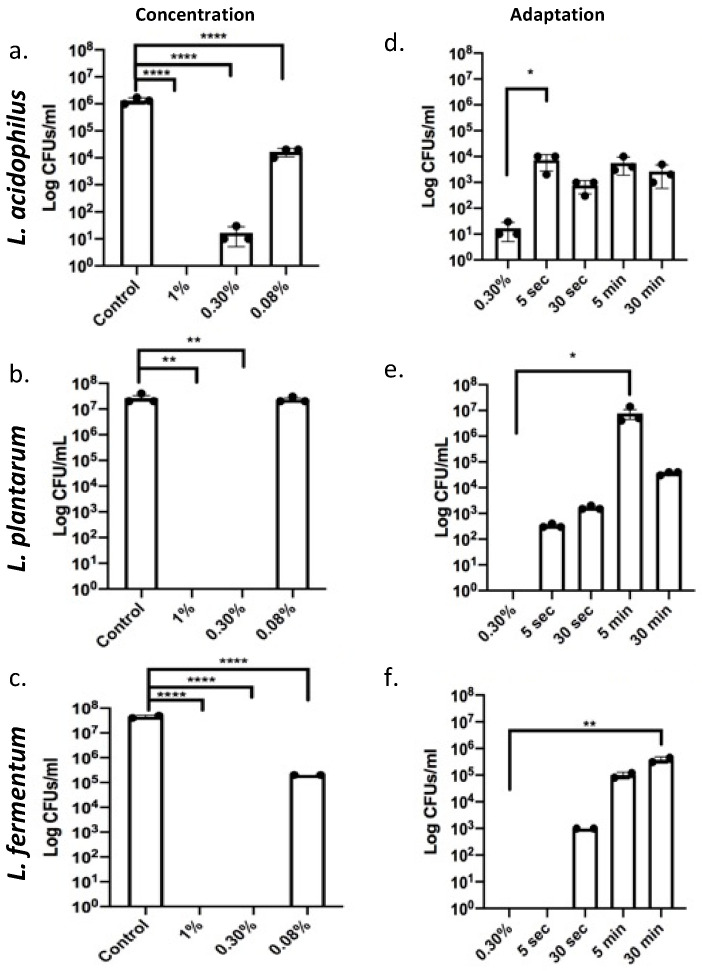
*Lactobacilli* adapt to cholate/deoxycholate bile salt challenge. (**a**–**c**) *Lactobacillus* species were exposed to 1% (positive control for killing), 0.30% (“lethal dosage”), and 0.08% (“sub lethal dosage”) during exponential phase for 30 s. *L. fermentum* and *L. plantarum* only tolerated 0.08% bile, *L. acidophilus* showed a decrease in growth after 0.30% bile exposure but tolerated 0.08%. **** *p* < 0.0001. (**d**–**f**) All *Lactobacillus* species were pre-exposed to the 0.08% “sub lethal dosage” for 5 s, 30 s, 5 min, or 30 min prior to being exposed to 0.30% “lethal dosage” for an additional 30 s. A total of 0.30% non-adapted *Lactobacilli* are used as a control for comparisons. Increased CFUs/mL upon sub lethal dosage exposure indicate adaption. *L. acidophilus* and *L. plantarum* adapted even after 5 s pre-exposure and survived the otherwise lethal 0.03% bile. *L. fermentum* required a 30 s pre-exposure * *p* < 0.05, ** *p* < 0.01, **** *p* < 0.0001.

**Figure 2 antioxidants-12-01314-f002:**
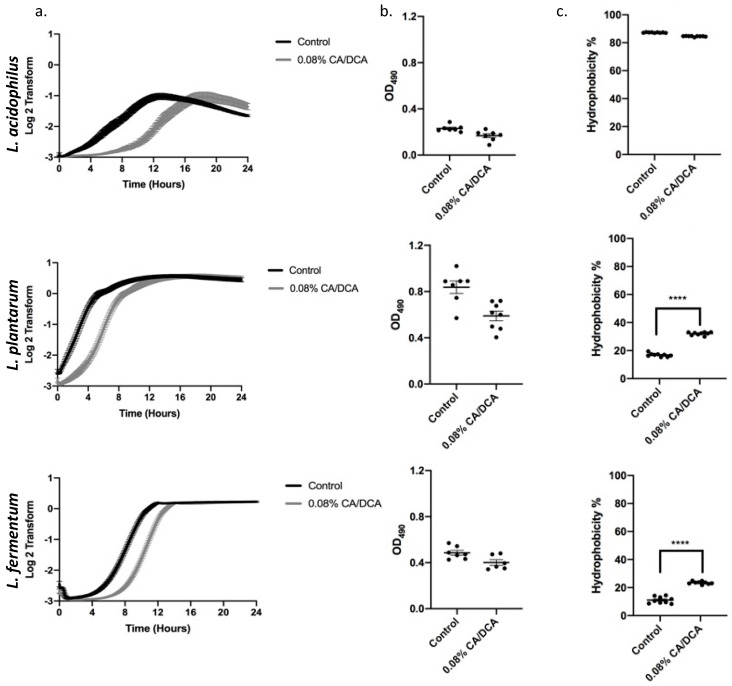
Non-lethal cholate/deoxycholate (CA/DCA) exposure delays exponential growth phase entry without the induction of a stress response. (**a**) Growth curves for three *Lactobacillus* strains were determined after 30 s pre-exposure to 0.08% CA/DCA. CA/DCA pre-exposure delays growth of *Lactobacilli* initially during recovery, but all three *Lactobacilli* reached the stationary phase comparable to the untreated control. (**b**) The 24 h exposure to 0.08% CA/DCA does not increase biofilm formation. (**c**) *L. acidophilus* hydrophobicity remained high while *L. plantarum* and *L. fermentum* increased after CA/DCA exposure. **** *p* < 0.0001.

**Figure 3 antioxidants-12-01314-f003:**
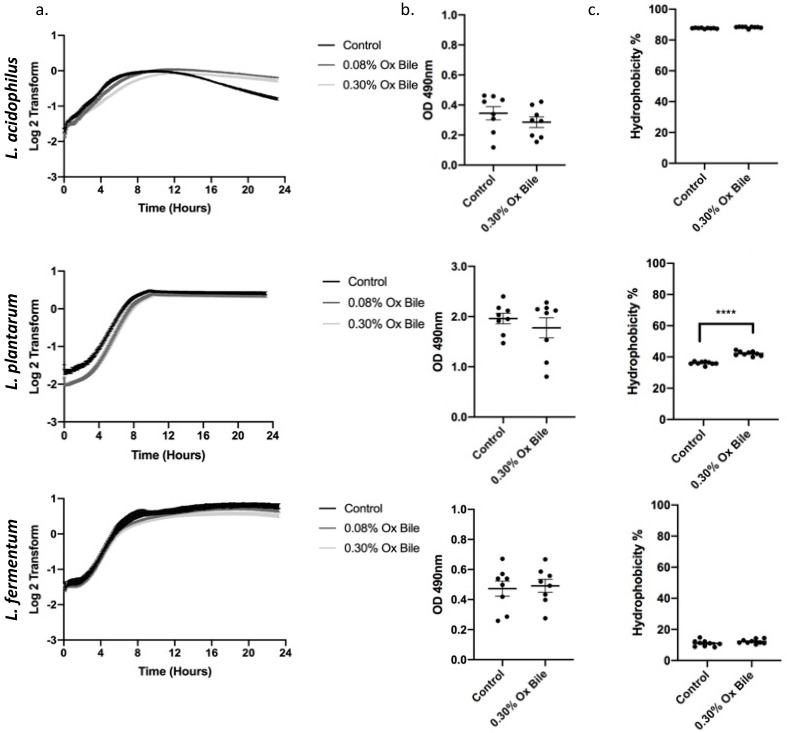
Ox bile as a physiological model for refluxate shows a delay on *Lactobacilli* growth without a stress response. (**a**) Growth curves for three *Lactobacillus* spp. were measured for 24 h during constant exposure to 0.30% and 0.08% ox bile. There is a delay in growth for *L. acidophilus and L. plantarum* before recovery and entry to the stationary phase. (**b**) Bacteria were grown for 24 h in the presence of 0.30% ox bile and stained with safranin to detect biofilm formation. No significant differences for a biofilm-related stress response were detected. (**c**) *L. plantarum* hydrophobicity is significantly increased after ox bile exposure, while *L. acidophilus* and *L. plantarum* remain unchanged. **** *p* < 0.0001.

**Figure 4 antioxidants-12-01314-f004:**
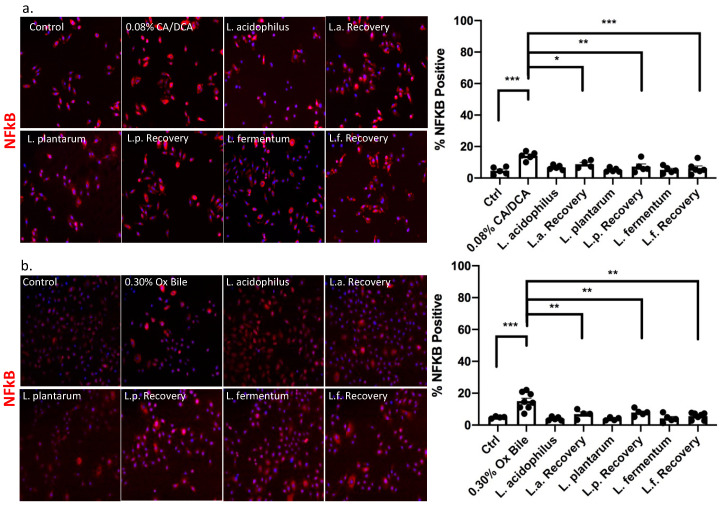
*Lactobacilli* mediate an anti-inflammatory response upon bile exposure. Esophageal epithelial cells were exposed to (**a**) CA/DCA and (**b**) ox bile for 30 s before a 30 min recovery period in the absence and presence of *Lactobacilli*. NFkB signal was detected with an Alexa 594-secondary antibody (red color) and quantified for each condition. The graph shows the number of positive cells per field. The NFkB signal was diminished when esophageal epithelial cells were co-cultured with *Lactobacilli* during the recovery compared single culture. * *p* < 0.05, ** *p* < 0.01, *** *p* < 0.001.

**Figure 5 antioxidants-12-01314-f005:**
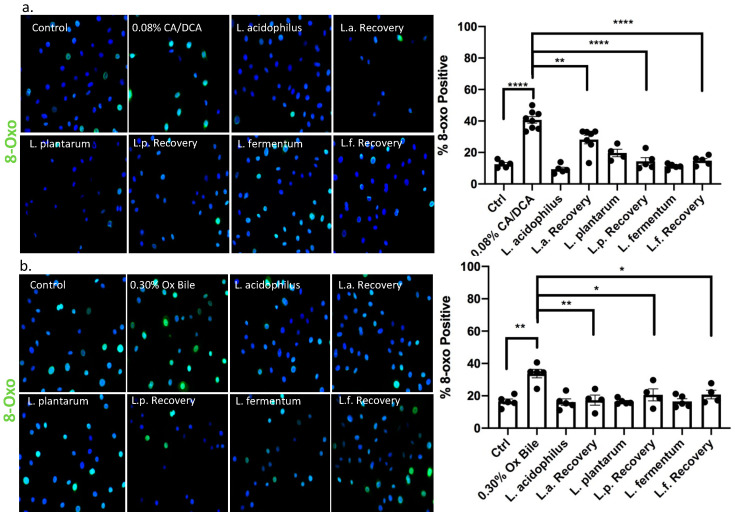
8-oxo guanine signal, a direct measure of bile acid-mediated DNA damage, is reduced in the presence of *Lactobacilli*. Esophageal epithelial cells were exposed to (**a**) CA/DCA or (**b**) ox bile known to induce ROS. Bile induces 8-oxoguanine DNA damage via ROS in epithelial cells alone as shown by immunofluorescence staining with antibody detecting 8-oxoguanine (8-oxo-G)-positive cells using an Alexa488-conjugated secondary antibody (green), but co-culture with *Lactobacilli* during the recovery time shows accelerated DNA repair. * *p* < 0.05, ** *p* < 0.01, **** *p* < 0.0001.

**Figure 6 antioxidants-12-01314-f006:**
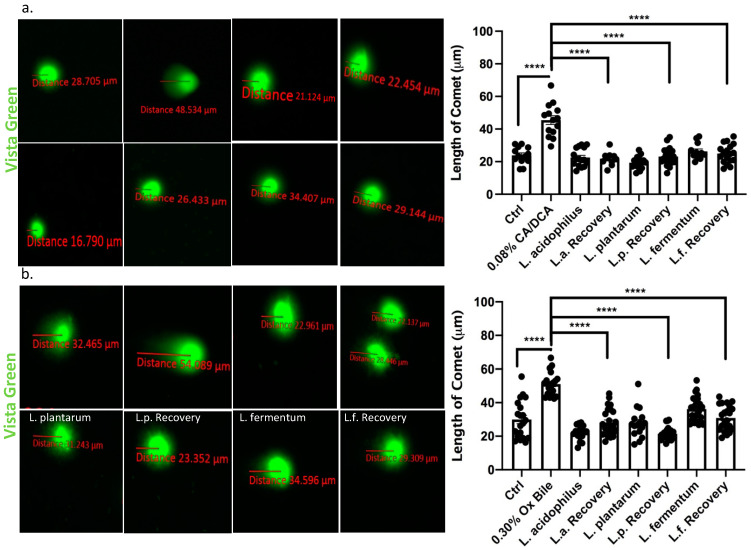
*Lactobacilli* accelerate DNA damage repair in response to bile in esophageal epithelial cells. Vista Green Dye was used to assess fragmented DNA in immortalized squamous esophageal epithelial cells after 30s CA/DCA (**a**) or ox bile (**b**) exposure. The distance between the COMET head and tail was measured as indicated. Epithelial cells co-cultured with *Lactobacilli* during the recovery time show accelerated DNA repair as evident by a shorter COMET length. **** *p* < 0.0001.

**Figure 7 antioxidants-12-01314-f007:**
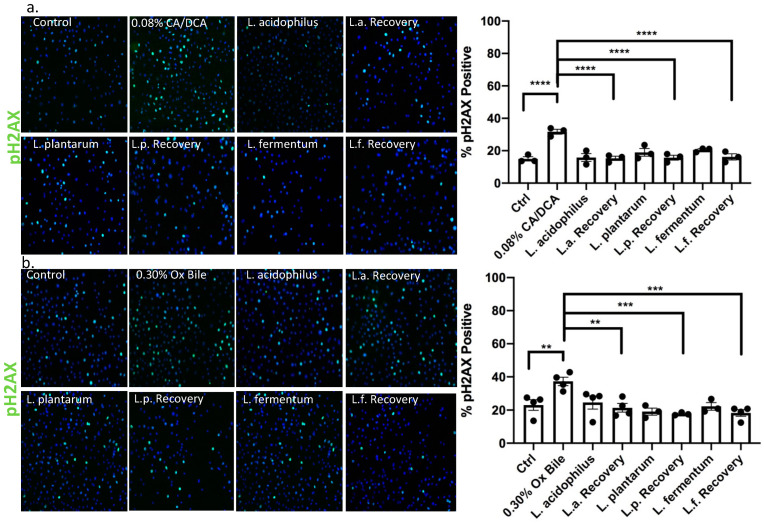
*Lactobacilli* induce DNA damage repair by the activation of pH2AX in response to bile in esophageal epithelial cells. Esophageal epithelial cells after 30 s exposure to (**a**) CA/DCA or (**b**) ox bile show a signal for phosphorylated H2AX (pH2AX) after 30 min recovery as shown by immunofluorescence using pH2AX antibody and Alexa488-conjugated secondary antibody (green). Epithelial cells co-cultured with *Lactobacilli* during the recovery time show accelerated DNA repair, demonstrating a comparable signal for pH2AX as untreated control. ** *p* < 0.01, *** *p* < 0.001, **** *p* < 0.0001.

**Figure 8 antioxidants-12-01314-f008:**
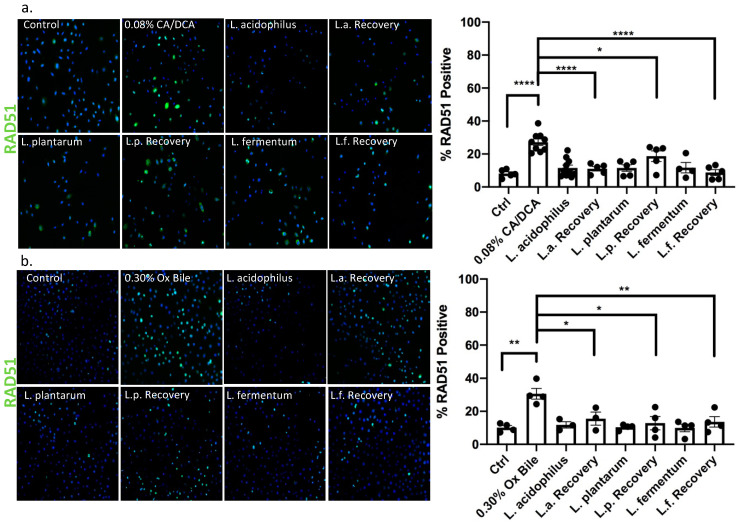
*Lactobacilli* exert RAD51 nuclear location and assistance in homologous repair function. Immunofluorescence with RAD51-specific antibody and Alexa488-conjugated secondary antibody (green) 30 min after a 30 s exposure to (**a**) CA/DCA or (**b**) ox bile shows an increase in positive esophageal epithelial cells. Epithelial cells co-cultured with *Lactobacilli* during the recovery time have a RAD51 positive signal that is similar to the untreated control. * *p* < 0.05, ** *p* < 0.01, **** *p* < 0.0001.

## Data Availability

Data is contained within this article and supplementary material.

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
