# Peer review of "Lactobacillus sp. Facilitate the Repair of DNA Damage Caused by Bile-Induced Reactive Oxygen Species in Experimental Models of Gastroesophageal Reflux Disease"

_antioxidants, 2023, doi:10.3390/antiox12071314_

Round 1

Reviewer 1 Report

The manuscript 2426051 entitled “Lactobacillus spec. facilitate the repair of DNA damage caused by bile-induced reactive oxygen species in experimental models of gastroesophageal reflux disease” reports the protection ability of three different Lactobacillus species, namely L. acidophilusL. plantarumL. fermentum (see below my comments on the recent taxonomy of Lactobacillus) against oxidative stress using a model of the gastroesophageal reflux disease. The approach used seems appropriate and data sounds scientific. Below are the recommended alterations in order to accept the manuscript to publication.

1.     Please review the taxonomy of Lactobacillus here: Zheng J, Wittouck S, Salvetti E, et al. A taxonomic note on the genus Lactobacillus: Description of 23 novel genera, emended description of the genus Lactobacillus Beijerinck 1901, and union of Lactobacillaceae and LeuconostocaceaeInt J Syst Evol Microbiol. 2020. doi: 10.1099/ijsem.0.004107.

2.     Title: use the abrreviation sp. instead of spec. The same in line 62.

3.     Line 13 and all over the manuscript: all scientific names are in italic.

4.     Line 69 change strains to species.

5.     Line 98 after Begley indicate the reference number.

6.     Line 100 how 0.08 % is a sub-lethal dose, 0.3% the lethal dose and 1% is the negative control?

7.     Line 103 change to time intervals

8.     Line 115 Indicate the number of biological and technical replicates?

9.     Line 127 the blanks consisted of what? Again, the number of biological and technical replicates are required.

10.  Line 135 change to OD600nm

11.  Line 145 media? you mean medium?

12.  Line 151 what media? perhaps medium?

13.  Line 162 change to (section 2.7)

14.  Line 177 What was the Statistical significance value considered? and when the analysis was statistically significant, what post hoc test was performed?

15.  Line 188 indicate the specie: monocytogenes

16.  Line 205-206  graphs of Figure 1: What is missing in the graphs B,D and F is the survival of the non-adapted bacterial cultures for the same time intervals. New graphs are required.

17.  Legend of Figure 1: rewrite this legend is confused.

18.  Line 207 and not E?

19.  Line 218-222 For the support of this statement I recommend to cite this article: Multicellular and unicellular responses of microbial biofilms to stress. Daniel K.H. Rode et al Biological Chemistry

https://doi.org/10.1515/hsz-2020-0213 or any other paper that stand this.

20.  Line 222 measured or observed?

Line 223 -233 rearrange the mention of the graphs; this is not clear. Indicate the significant differences (or none).

21.  Line 224 Reference 20 use a more recent reference, there are plenty!

22.  Line 245 for Figure 3B if there are no significant differences show this.

23.  Line 246 I do not agree: with CA/DCA, L. acidophilus shows a lag phase of 8 h and L. fermentum a lag phase of about 5 h. The growth curves in the presence of CA/DCA and ox bile are not similar!

24.  Line 251-256 Also rewrite the legend of Figure 3.

25.  Line 269 after 0.30% ox bile add for 30 seconds before a 30-minute recovery period

26.  Line 277-283 maybe indicate the 2 panels with A and B?

27.  Line 279-283. The images need an appropriate explanation: the meaning of each colour.

28.  Line 305 red arrows are not visible. And the same as for Figure 4.

29.  Line 351-355 the same comments as for Figure 4 and 5.

30.  Line 357-360 the same comments as for Figure 4, 5 and 7.

31.  Line 367 Not so recent publication one of them has already 19 years old and another 10. Please rephrase.

32.  these reagents? or these conditions? 

33.  Line 416-429 as this was not explored in the study eliminate or briefly mention it.

34.  Line 440 rephrase; this is just a study.

35.  Line 444 a reference is required here to support the statement.

36.  Line 483-486 this does not seem to be in the context of the study. Why mix here the action of gram negative and gram positive bacteria?

37.  Line 501 regarding reference 93 I strongly believe that the authors will find a much more recent reference about the impact of Bifidobacterium in human health.

Author Response

We would like to thank the reviewer for the careful reading of the manuscript and the suggestions and comments to modify the text and data presentation for clarification and accuracy.

We corrected the taxonomy, abbreviations and formatting of italics for scientific names (Points 1-4) throughout the manuscript.

Line 98 we referred to (16) right after mentioning Begley.

We modified the materials and methods, the legend and text to explain the adaptation process. We also corrected the former “negative control” to be a positive control (for bacterial cell death). This included clarification of the procedure and the time points selected.

We also expanded the section to include more detail on the experiment in the text and Legend:

Pre-exposure with 0.08% CA/DCA for 5 sec, 30 sec, 5 min and 30 minutes allowed L. acidophilus and L. plantarum to adapt and survive the otherwise lethal exposure of 0.30% CA/DCA. L. fermentum required a pre-exposure of at least 30 sec for adaption.

Materials and Methods were corrected to include that experiments were performed as technical triplicates and repeated twice under the sections requested by the reviewer (line 115, line 129) in addition to the text in the statistics section (line 177).

Line 128 also explains now that blank is media only.

Line 177: We used a one-Way ANOVA when comparing multiple groups or conditions. For a statistically significant ANOVE, analysis was followed by a Dunnett’s correction when multiple groups/conditions were compared to controls. This information is expanded upon in the Materia’s and Methods section for statistics.

  1. Line 205-206 graphs of Figure 1: What is missing in the graphs B,D and F is the survival of the non-adapted bacterial cultures for the same time intervals. Line 201 and beyond in the text include now more detail on the experimental design in hopes to clarify the existing graphs: Based on the experiments by Begley we refer to, we show that 0.30% bile leads to an inhibition of growth for L. acidophilus and kills L. plantarum and L. fermentum. The 0.30% is therefore the non-adapted control in panels d-f.No untreated controls as in graphs a-c) were included. Instead the comparisons were made between pre-exposure with 0.08% CA/DCA for 5 sec, 30 sec, 5 min and 30 minutes and lethal 0.30% CA/DCA. If the reviewer is asking why we didn’t expose 0.30% non-adapted bacteria also for 5 sec, 30 sec, 5 min and 30 minutes, the reason is that the 30sec exposure reduced the survival for L. acidophilus or killed L. plantarum and L. fermentum as shown in a-c. We expanded the Materials and Methods and text to clarify this problem, now 194-209.

We named the figure panels in the text throughout the manuscript and made the labelling clearer (e.g., a-c, d-ef).

Text and legend for Figure 3 were modified according to the reviewer’s suggestions to be more explicit about the delay in growth and significant differences. Line 254 and beyond.

Text and Legend for Figure 4 give more detail on the procedure now and the labelling of the figure was edited according to the reviewer’s suggestion (a and b). The legend also refers to the Alexa594-antibody giving it the red signal. This was also addressed in Figures 5 and 7.

The discussion was altered to correct conditions for reagents, shorten the paragraph about APE1. Given its relevance to redox biology and Barrett’s tumorigenesis, we still highlight its function in redox signaling but with less emphasis. If the reviewer doesn’t share that opinion, we will, of course, delete this paragraph.

Similarly, while gram-negative and gram-positive bacteria have different interactions with ‘host’ cells, we felt that in the absence of a mechanism underlying the effects of probiotic Lactobacilli on esophageal epithelial cells, it is important to point to other reports that investigated the consequences of such interactions on DNA repair. If the reviewer doesn’t share that opinion, we will, of course, delete this paragraph.

Language:

Line 135 change to OD600nm now Line 138

Medium was changed to media

change to (section 2.7) now Lines 160 and 165

Line 189 indicate the specie: monocytogenes; we replaced with Salmonella sp.

Line 222 measured or observed? Measured

these reagents? or these conditions? Conditions, thank you.

Line 440 rephrase; this is just a study. Now Line 511, thank you!

References have been updated as requested and the text changed to delete ‘recent’ publications.

Reviewer 2 Report

This manuscript submitted to Antioxidants assessed the probiotic potential of Lactobacilli in bile-induced -injured esophageal cells, and therefore in gastroesophageal reflux disease. There was promising clinical data but this in vitro study is totally innovative. The authors used three strains of Lactobacilli: L. acidophilus, L. plantarum, L. fermentum. They firstly showed that the 3 strains acquired tolerance to lethal 0.30% CA/DCA bile salt exposure. All three Lactobacilli strains did not show significant alteration of bacterial growth and biofilm formation after 0.08% CA/DCA preexposure. CA/DCA preexposure did not decrease hydrophobicity, and even increased the hydrophobicity of L. plantarum and L. fermentum. Concordant results were obtained following ox bile preexposure (similar to human bile composition). The 3 strains inhibited esophageal epithelial cell NF-kB translocation induced by CA/DCA or ox bile, showing therefore anti-inflammatory properties. Anti-oxidative properties were also demonstrated using 8-oxo-G IHC assay. Then the authors showed the ability of Lactobacilli to repair DNA damage induced by BA, through recruitment of pH2AX/RAD51.

This is a straight forward manuscript, very well written, easy to understand, very comprehensive. No details are missing, from the introduction to the conclusion. The choice of methodologies is good, no controls are lacking. No overstatement.

Minor comments

Figure 1B-F. Only one preexposure time seems to induce significant tolerance. Is the p-value for other preexposure times not statistically different from the control (0.30%)?

Lines 240-244. The choice of ox bile is well explained but not the 0.30% concentration. Is it physiologically relevant?

Figure 4: Are the p-value indicated for the comparison between the 0.08% CA/DCA (or 0.30% ox bile) group and each Lactobacillus recovery group? A single line for the 3 Lactobacillus recovery groups is confusing. Same comment for figures 5, 6, 7,8.

Fig. S1 and S2 do not appear in the text.

Fig S1: What is the meaning of “constant”?

Author Response

We would like to thank the reviewer for the positive feedback and the suggestions to improve the manuscript.

Figure 1 B-F: We added more information to the text and materials and methods to explain the procedure. When comparing multiple groups, a One-ay ANOVA was used to calculate significance. A significant ANOVA (which we found for all the pre-exposure time points in the graph) was followed by Dunnett’s correction where multiple groups were compared. This additional calculated resulted in only one of the pre-exposure conditions to be significant with that particular analysis.

We added a new reference (25) for the range of human bile to the introduction for the ox bile physiology at 0.3% at line 250.

Figures 4, 5, 6, 7,8. Comparisons between each recovery group and the control were significant. We changed the comparison line in all the figures to make that clear.

The Figure legend for S2 was altered to give more information on the treatments: incubation with CA/DCA was for 5 min, 30 min or with bile present for the entire 48 hours experiment (constant).

The text includes a reference to the supplemental figures at

Line 279 for Figure S1 and line 364 for Figure S2.